# Photoinhibition of Photosystem I Induced by Different Intensities of Fluctuating Light Is Determined by the Kinetics of ∆pH Formation Rather Than Linear Electron Flow

**DOI:** 10.3390/antiox11122325

**Published:** 2022-11-24

**Authors:** Qi Shi, Xiao-Qian Wang, Zhi-Lan Zeng, Wei Huang

**Affiliations:** 1Kunming Institute of Botany, Chinese Academy of Sciences, Kunming 650201, China; 2University of Chinese Academy of Sciences, Beijing 100049, China

**Keywords:** electron transport, fluctuating light, photosystem I, photoinhibition, proton gradient

## Abstract

Fluctuating light (FL) can cause the selective photoinhibition of photosystem I (PSI) in angiosperms. In nature, leaves usually experience FL conditions with the same low light and different high light intensities, but the effects of different FL conditions on PSI redox state and PSI photoinhibition are not well known. In this study, we found that PSI was highly reduced within the first 10 s after transition from 59 to 1809 μmol photons m^−2^ s^−1^ in tomato (*Solanum lycopersicum*). However, such transient PSI over-reduction was not observed by transitioning from 59 to 501 or 923 μmol photons m^−2^ s^−1^. Consequently, FL (59-1809) induced a significantly stronger PSI photoinhibition than FL (59-501) and FL (59-923). Compared with the proton gradient (∆pH) level after transition to high light for 60 s, tomato leaves almost formed a sufficient ∆pH after light transition for 10 s in FL (59-501) but did not in FL (59-923) or FL (59-1809). The difference in ∆pH between 10 s and 60 s was tightly correlated to the extent of PSI over-reduction and PSI photoinhibition induced by FL. Furthermore, the difference in PSI photoinhibition between (59-923) and FL (59-1809) was accompanied by the same level of linear electron flow. Therefore, PSI photoinhibition induced by different intensities of FL is more related to the kinetics of ∆pH formation rather than linear electron flow.

## 1. Introduction

Plants use light energy to drive photosynthetic electron flow and CO_2_ fixation. Light intensity usually fluctuates in nature, and this condition of light is called fluctuating light (FL) [1]. During FL, a sudden increase in illumination induces an immediate increase in the electron flow of photosystem II (PSII) [2,3]. Concomitantly, the CO_2_ fixation needs much more time to be fully activated [4,5]. Such an imbalance of photosynthetic electron flow and CO_2_ fixation restricts the electron sink downstream of photosystem I (PSI), and the resulting PSI over-reduction would trigger the production of reactive oxygen species (ROS) in PSI [6,7]. Since ROS produced in PSI cannot be immediately scavenged by the antioxidative system [8], FL can induce preferential photoinhibition of PSI in many angiosperms, such as *Arabidopsis thaliana* [6,9,10,11,12], tobacco (*Nicotiana tabacum*) [13,14], rice (*Oryza sativa*) [15], tomato (*Solanum lycopersicum*) [16,17], and *Bletilla striata* [2,18]. Owing to the importance of PSI in photosynthetic electron flow and photoprotection, PSI photoinhibition induced by FL can depress photosynthetic efficiency and thus impairs plant growth [19,20,21,22,23,24]. Therefore, response of PSI to FL is critical to plant grown in natural field conditions.

When plants experience FL, the primary reason for PSI photoinhibition is the over-reduction in PSI electron carriers, and PSI redox state is modulated at the donor and/or acceptor side [6,10]. In acceptor side regulation, excess excitation energy in PSI is consumed by alternative electron flows meditated by flavodiiron proteins (FDPs) [7,25,26,27,28,29] or the water–water cycle [30,31,32,33]. From an evolutionary perspective, FDPs are conserved in cyanobacteria [25], green algae [26], liverworts [34], mosses [7,27,28], ferns [35], and gymnosperms [29] but are lost in angiosperms during evolution [36]. Furthermore, the water–water cycle just exists in some specific groups of angiosperms, and the activity is species-dependent and sensitive to environmental temperature [30,37,38]. Therefore, acceptor-side regulation is not a major strategy used by angiosperms to cope with FL. Alternatively, donor-side regulation mediated by proton gradient (ΔpH) across the thylakoid membranes is essential for PSI photoprotection in angiosperms [39,40,41,42]. With the increase in ΔpH, the downregulation of plastoquinol oxidation in Cytochrome *b6f* complex slows down the electron flow from PSII and thus alleviates PSI over-reduction. Once ΔpH formation was disturbed in *Arabidopsis pgr5*, *cfq,* and *hope2* mutants, they displayed PSI over-reduction under high light and severe PSI photoinhibition under FL [6,9,11,43,44]. Upon transition from very low light to extreme high light, most angiosperms cannot generate a sufficient ΔpH within the initial seconds, which increases the risks of transient PSI over-reduction and subsequent PSI photoinhibition [3,18,45,46,47]. Therefore, the susceptibility to PSI photoinhibition is related to the kinetics of ΔpH formation after any increase in light.

Generally, ΔpH formation is influenced by light intensity under optimal conditions. With the increase in light intensity, both increases in linear electron flow (LEF) and cyclic electron flow (CEF) around PSI upregulate the influx activity of H^+^ from stroma to the lumen of thylakoid, contributing to the increase in ΔpH [40,43,48]. By transitioning to the same high light, a higher background low-light phase facilitates ΔpH formation within the first seconds and thus alleviates PSI over-reduction [49]. As a result, the intensity of background low light significantly affects the extent of FL-PSI photoinhibition. In nature, light intensity might shift from the same low light to different intensity of high light. However, it is still unclear what intensity of high light irradiance will cause PSI photoinhibition in FL conditions with the same low light. In WT and *pgr5* mutant of *Arabidopsis thaliana*, an increase in the intensity of high light phases did aggravate the extent of FL-induced PSI photoinhibition. These observations could be explained if ΔpH formation is more inadequate upon a sudden transition to a stronger light intensity.

PSI photoinhibition occurs only when the electron transport flow to PSI exceeds the capacity of PSI electrons downstream. Under moderate PSII photoinhibition, the downregulation of PSII electron flow decreases the excitation pressure to PSI upon transition from low to high light [50]. In the *pgr5* mutant of *Arabidopsis thaliana*, a minimal activity of the oxygen-evolving complex prevented PSI over-reduction under high light and thus rescued the death of *pgr5* mutant grown under FL [51]. In shade-adapted plant *Paris polyphylla*, PSI over-reduction under FL did not occur, owing to the relatively low capacity of LEF [46]. In tomato plants with low leaf nitrogen content, the relatively low capacity of LEF prevented the PSI over-reduction under FL and thus led to the insusceptibility to PSI photoinhibition [17]. Therefore, the capacity of LEF is another important factor affecting the extent of FL-induced PSI photoinhibition. According to the model of the light response curve, LEF gradually increases with the increasing irradiance and becomes stable at the light saturating point. Under FL conditions with the same low light and different high light intensities, the extent of FL-induced PSI photoinhibition might be related to the change in LEF after the light transition.

In this study, we measured PSI, PSII, and electrochromic shift signal in tomato leaves illuminated under FL conditions with the same low light and different intensities of high light illumination. The major aim is to elucidate the following: (1) Does the intensity of high light significantly affects FL-induced PSI photoinhibition? (2) Is the extent of FL-induced PSI photoinhibition mainly determined by the kinetics of ∆pH or the response of LEF? We found that FL-induced PSI photoinhibition was aggravated upon the transition from the same low light to a stronger high light, which was related more to the kinetics of ∆pH formation than to the response of LEF.

## 2. Materials and Methods

### 2.1. Plant Material and Growth Condition

We used tomato (*Solanum lycopersicum* Miller cv. Hupishizi) plants in this study. Plants were grown in a greenhouse with day/night air temperatures of 30/20 °C and a relative air humidity of 60–70%. The light condition was controlled by a non-woven textile cover over them, reducing the photosynthetic active radiation (PAR) to approximately 40% of the sunlight (measured by a Li-1400 datalogger, Li-Cor Biosciences, Lincoln, NE, USA). As a result, the maximum light intensity exposed to leaves at noon was approximately 800 μmol photons m^−2^ s^−1^. All plants were grown with humus soil with an initial soil N content of 2.1 mg/g and fertilized with Peters Professional’s water solution. To avoid any water and nutrient stress, plants were watered and fertilized every day. After cultivating for a month, mature but not senescent leaves with the maximum photosynthetic capacity of the whole plant in elongation stage were used for photosynthetic measurements.

### 2.2. Chlorophyll Fluorescence and P700 Measurement

We used a Dual-PAM 100 measuring system (Heinz Walz, Effeltrich, Germany) to measure PSI and PSII parameters under atmospheric CO_2_ conditions at 25 °C. After 15 min dark adaptation, PSI and PSII activities were measured by a saturating pulse (20,000 μmol photons m^−2^ s^−1^, 300 ms). Subsequently, tomato leaves were illuminated at 1455 μmol photons m^−2^ s^−1^ for 5 min to activate photosynthetic electron sinks. Subsequently, leaves were exposed to FL alternating between low light (59 μmol photons m^−2^ s^−1^, 2 min) and high light (501, 923 or 1809 μmol photons m^−2^ s^−1^, 1 min) for eight cycles in total. PSI and PSII parameters were recorded simultaneously during these FL treatments, and after that, *P_m_* was measured to evaluate the extent of PSI photoinhibition.

The chlorophyll fluorescence parameters were calculated as follows: the quantum yield of PSII photochemistry, Y(II) = (*F_m_′* − *F_s_*)/*F_m_′*; non-photochemical quenching in PSII, NPQ = (*F_m_* − *F_m_′*)/*F_m_′*; the quantum yield of non-regulatory energy dissipation in PSII, Y(NO) = *F_s_*/*F_m_*. *F_m_* and *F_m_′* are the maximum fluorescence intensity after dark and light acclimation, respectively. *F_s_* is the light-adapted steady-state fluorescence. The PSI photosynthetic parameters were measured based on P700 signal. PSI parameters were calculated as follows: the quantum yield of PSI photochemistry, Y(I) = (*P_m_′* − *P*)/*P_m_*; the quantum yield of PSI non-photochemical energy dissipation due to donor-side limitation, Y(ND) = *P*/*P_m_*; the quantum yield of PSI non-photochemical energy dissipation due to acceptor side limitation, Y(NA) = (*P_m_* − *P_m_′*)/*P_m_*. The photosynthetic electron transport rate was calculated as ETRI (or ETRII) = PPFD × Y(I) [or Y(II)] × 0.84 × 0.5, where PPFD is the photosynthetic photons flux density, 0.84 is the assumed light absorption of incident irradiance, and 0.5 is the fraction of absorbed light reaching PSI or PSII. We calculated the relative CEF by subtracting ETRII from ETRI.

### 2.3. Electrochromic Shift (ECS) Measurement

We used a Dual-PAM 100 equipped with a P515/535 emitter-detector module to measure the ECS signals [52]. After dark adaptation for 15 min, the 515 nm absorbance change induced by a single turnover flash (ECS_ST_) was measured. Leaves were then illuminated at 1455 μmol photons m^−2^ s^−1^ for 5 min for light adaptation, followed by illumination at 59 μmol photons m^−2^ s^−1^ for 2 min. Afterwards, light intensity was changed to 501, 923 or 1809 μmol photons m^−2^ s^−1^, respectively, and ECS dark interval relaxation kinetics (DIRK_ECS_) were recorded after this light transition for 10 s or 60 s according to the method previously reported. The total proton motive force (*pmf*), the proton gradient (ΔpH) component of proton motive force, and the chloroplast ATP synthase activity (*g*_H_^+^) were estimated from the DIRK_ECS_ [53,54,55,56,57].

### 2.4. Statistical Analysis

All data are presented as mean values of five leaves from five independent plants. One-way ANOVA and *t*-tests were used to determine whether significant differences existed between different treatments (*α* = 0.05). We used the software SigmaPlot 10.0 (Chicago, IL, USA) for graphing and fitting.

## 3. Results

### 3.1. Dynamic Changes in PSI and PSII Redox State under Fluctuating Light

We first measured the dynamic changes in PSI and PSII redox state under FL conditions with the same low light (59 μmol photons m^−2^ s^−1^) and different high light irradiance (501, 923 or 1809 μmol photons m^−2^ s^−1^). Upon a sudden transition from low to high light, the higher intensity of high light irradiance was coupled with the lower quantum yield of PSI photochemistry [Y(I)] (Figure 1A). Moreover, the values of Y(ND) (the donor-side limitation) and Y(NA) (the acceptor-side limitation) were remarkably affected by the intensity of high light phases. Y(ND) and Y(NA) reflected the redox state of PSI which was closely associated with the extent of PSI photoinhibition. Within the first 10 s after transition from low to high light, tomato leaves showed a lower level of Y(ND) and a higher level of Y(NA) in FL (59-1809) than in FL (59-501) and FL (59-923) (Figure 1B,C), indicating a severe over-reduction in electron carriers in FL (59-1809). Such over-reduction of PSI was gradually alleviated during further exposure. As a result, a higher intensity of high light phase inevitably aggravated the transient PSI over-reduction in FL.

Similar to Y(I), the value of quantum yield of PSII photochemistry [Y(II)] in high light phases in FL (59-1809) was lower than FL (59-501) and FL (59-923) (Figure 2A). After light intensity increased abruptly, non-photochemical quenching (NPQ) rapidly increased to the steady-state value in 1 min, with the highest value in FL (59-1809) (Figure 2B). Consequently, the tomato leaves displayed similar non-regulated energy dissipation in PSII [Y(NO)] when the light intensity changed from 59 to 501, 923 or 1809 μmol photons m^−2^ s^−1^ (Figure 2C). A high Y(NO) indicated that the excess excitation energy in PSII could not be fully dissipated harmlessly as heat through NPQ, implying the accumulation of excess excitation in PSII electron carriers. These results demonstrated that no significant difference in the over-reduction of PSII was observed in FL treatments with the same low light and different high light intensities, which was unlike that in PSI.

During the fluctuating light treatments, electron transport rates through PSI and PSII (ETRI and ETRII) during low light phases did not differ significantly among these three FL treatments (Figure 3A,B). Consequently, all FL treatments displayed nearly the same low levels of ETRI−ETRII under low light (Figure 3C). During high light phases, ETRI values were highest in FL (59-1809), followed by FL (59-923) and FL (59-501) (Figure 3A). Concomitantly, the ETRII values during high light phases were similar in FL (59-1809) and FL (59-923), which were much higher than in FL (59-501) (Figure 3B). As a result, FL (59-1809) displayed the highest value of ETRI−ETRII, followed by FL (59-923) and FL (59-501) (Figure 3C). In FL (59-501) and FL (59-923), the value of ETRI−ETRII rapidly increased to a peak in 10 s and then slowly decreased to a steady state after transitioning to high light. By contrast, the value of ETRI−ETRII gradually climbed to the maximum in 30 s and then decreased to a little low level in FL (59-1809) during the first five cycles of FL treatment. Since ETRI comprises linear electron flow, the water–water cycle, and cyclic electron flow, and ETRII only comprises linear electron flow and the water–water cycle, a high level of ETRI−ETRII is an indicator of CEF activation, which was supported by the low value of ETRI−ETRII at low light and the increased value under high light. Therefore, we may safely draw the conclusion that the stimulation extent of CEF under FL was affected by the background high light irradiance.

PSI photoinhibition can be manifested by the decrease in the maximum photo-oxidizable P700 (*P_m_*). Therefore, we measured the values of *P_m_* before and after FL treatments to evaluate the extent of FL-induced PSI photoinhibition. After FL treatments, *P_m_* decreased by 15%, 7%, and 6% in FL (59-1809), FL (59-923), and FL (59-501), respectively (Figure 4A). As a result, the intensity of the high light phase had a significant impact on FL-induced PSI photoinhibition. Moreover, further experiments indicated that the losing PSI activity under FL was positively correlated to the average Y(NA) within the first 10 s after switching to high light (Figure 4B). In FL alternating between 59 and 1809 μmol photons m^−2^ s^−1^, the higher Y(NA), meaning severer over-reduction in PSI, led to stronger PSI photoinhibition. 

### 3.2. Kinetics of Proton Gradient and Chloroplast ATP Synthase Activity and under Fluctuating Light

Due to the important roles of ∆pH and the activity of chloroplast ATP synthase (*g*_H_^+^) in regulating the redox state of PSI under high light, we examined the kinetics of ∆pH and *g*_H_^+^ after the transition from low to high light in these three FL conditions. In FL (59-501), no significant difference in ∆pH was observed between 10 s and 60 s after transitioning from low to high light. However, a significant difference in ∆pH was observed between 10 s and 60 s in the other two FL conditions, especially in FL (59-1809) (Figure 5A). Therefore, tomato plants need more time to fulfill the ∆pH formation when transitioning from the same low light to a stronger high light. After the transition to high light for 60 s, the ∆pH value was highest in FL (59-1809), followed by FL (59-923) and FL (59-501), indicating that the optimization of the PSI redox state at 1809 μmol photons m^−2^ s^−1^ required a relatively higher ∆pH formation. There was no meaningful difference in the value of *g*_H_^+^ between different FL treatments after the transition to high light for 10 s and 60 s (Figure 5B), showing that the efflux activity of H^+^ did not differ among these FL conditions.

### 3.3. Relationships between Proton Gradient and Photoinhibition under Fluctuating Light

Based on the data of PSI and ∆pH during FL treatments, we analyzed the relationships between the kinetics of ∆pH formation (∆pH_60 s_ − ∆pH_10 s_) and PSI photoinhibition induced by FL, and found that the value of ∆pH_60 s_ − ∆pH_10 s_ was tightly correlated to the average Y(NA)_10 s_ and the decrease in *P_m_* (Figure 6A,B). In FL (59-501) and FL (59-923), the relatively low values of ∆pH_60 s_ − ∆pH_10 s_ were accompanied by low levels of PSI over-reduction and PSI photoinhibition. A high level of ∆pH_60 s_ − ∆pH_10 s_ in FL (59-1809) was accompanied by a high value of Y(NA) and a drastic decrease in *P_m_*. The transient PSI over-reduction in FL (59-1809) was tightly correlated to an insufficient ∆pH (Figure A1). Therefore, the kinetics of ∆pH formation after the light transition is critical to photoprotection under FL. When exposed to FL with different intensities of high-light phases, the rapid and sufficient buildup of ∆pH within the first 10 s after transitioning to high light could prevent the transient PSI over-reduction and thus diminish the risk of PSI photoinhibition.

## 4. Discussion

In nature, leaves always experience rapid fluctuations of light intensity on timescales from seconds to minutes, owing to changes in leaf angle, cloud movement, forest gaps, and canopy cover [1]. When light intensity received by leaf decreases suddenly, it just causes a reduction in the CO_2_ assimilation rate without severe photoinhibition [58,59,60]. By contrast, upon a sudden increase in irradiance, the PSII electron flow increased rapidly and the superfluous electrons transported to PSI cannot be immediately consumed by primary metabolism, which could induce selective PSI photoinhibition in angiosperms [6,9,10,11,17,37]. Once PSI was photodamaged, the major electron transport flows, LEF and CEF, were restricted, followed by the suppression of CO_2_ assimilation and photoprotection, impairing plant growth [19,20,21,22,23,24]. Therefore, understanding the mechanisms of FL-induced PSI photoinhibition is of importance to improve plant growth. The intensity of FL is determined by both low and high light, implying that FL-induced PSI photoinhibition can be influenced by the intensities of low and/or high light. Decreasing the background low light could aggravate FL-induced PSI photoinhibition in *Bletilla striata* [17]. However, it is unclear how the intensity of high light irradiance affects FL-induced PSI photoinhibition. 

We here found that the intensity of high light irradiance can significantly affect PSI redox state and the extent of PSI photoinhibition during FL. PSI was transiently over-reduced in FL (59-1809), while such transient PSI over-reduction was not observed in FL (59–501) and FL (59-923) (Figure 1C). After eight cycles of treatment, FL (59-1809) caused stronger PSI photoinhibition than FL (59-501) and FL (59-923) (Figure 4A). It is generally accepted that PSI photoinhibition is largely due to oxidative damage, which rarely occurs under anaerobic conditions or a high level of PSI oxidation [7,8,11,41,51]. Consequently, O_2_ over-reduction and PSI over-reduction are two prerequisites of PSI photoinhibition. In our present study, it is also indicated that the transient and severe over-reduction in PSI after transitioning to high light is the primary cause of PSI photoinhibition induced by FL (Figure 4B). Therefore, avoiding extreme high light is a feasible strategy to diminish the detrimental effects of FL on crop photosynthesis.

In angiosperms, ∆pH is the key signal for regulating PSI redox state when faced with FL [11,39,40]. In *Arabidopsis pgr5* mutants with the impairment of CEF, the restriction of ∆pH formation under high light resulted in prolonged PSI over-reduction under high light, leading to severe PSI photoinhibition when exposed to FL [9,10,11]. After the transition from very low to extremely high light, plants usually cannot generate a sufficient ∆pH in the first 10 s [30,31,35]. According to the model of the light response curve, a higher light intensity needs a higher ∆pH to prevent PSI over-reduction under a steady state [61]. Therefore, upon transition from the same low light to a stronger high light, plants might need more time to fulfill the formation of a sufficient ∆pH. Such delaying of the formation of ∆pH would be the major cause of FL-induced PSI photoinhibition. To test this hypothesis, the kinetics of ∆pH formation in different FL conditions were measured. We found that tomato leaves could rapidly build up a sufficient ∆pH in FL (59-501) but failed in FL (59–923) and FL (59-1809) (Figure 5A). In further analysis, we observed that the delaying formation of ∆pH between 10 s and 60 s tightly affected PSI over-reduction and PSI photoinhibition in FL (Figure 6). Therefore, the kinetics of ∆pH formation plays a key role in determining the PSI performance under FL.

In addition to ∆pH formation, LEF is thought to be important for PSI redox state [62]. In the *Arabidopsis pgr5* mutant, a decrease in LEF upon moderate PSII photoinhibition could alleviate PSI over-reduction and protect PSI activity under high light [50]. When the activity of the oxygen-evolving complex was minimal in the *Arabidopsis pgr5* mutant, PSI over-reduction was not observed under high light, and the lethal phenotype under FL was rescued [51]. Low capacity of LEF prevented PSI over-reduction and PSI photoinhibition under FL in shade-adapted plants *Paris polyphylla* [46] and tomato plants grown with nitrogen deficiency [17]. Therefore, the rapid increase in LEF is the prerequisite of PSI over-reduction upon a sudden increase in irradiance. In our present study, no significant decrease in Y(II) was observed in these FL treatments (Figure 2), indicating that FL-induced PSII photoinhibition was very slight. Interestingly, no significant difference in LEF was observed between FL (59-923) and FL (59-1809) (Figure 3B), but severe transient PSI over-reduction just occurred in FL (59-1809) (Figure 1C). Furthermore, a higher LEF under high light in FL (59-923) than in FL (59-501) was accompanied by a similar extent of PSI photoinhibition (Figure 3B and Figure 4A). These results indicate that LEF plays a minor role in affecting PSI redox state and PSI photoinhibition under FL conditions with the same low light and different high light.

As we know, the Calvin–Benson cycle and photorespiratory pathway in plants needs the ATP/NADPH ratios of 1.5 and 1.75, respectively [63]. However, the ATP/NADPH production ratio produced by LEF is just 1.29, which is not enough for primary metabolism in vivo [55,64]. Consequently, one of the most important roles of CEF in sustaining photosynthesis is to provide additional ATP [65]. It is known to us that CEF also protects PSI against photoinhibition when faced with FL conditions [6,10,11,15]. In particular, CEF was highly stimulated after a quick transition to high light in *A. thaliana*, *Bletilla striata*, and *Cerasus cerasoides* [2,9,45]. In this case, the activation of CEF can help to rapidly generate ∆pH, and thus it will be beneficial for the photosynthetic control at the cytochrome *b6f* complex [66]. Moreover, it increases the ATP/NADPH production ratio within the chloroplast and facilitates primary metabolism [6]. We here achieved consistent results shown as the kinetics of ETRI−ETRII (Figure 3C) and ∆pH (Figure 5A). When the light irradiance transitioned from 59 to 501 or 923 μmol photons m^−2^ s^−1^, CEF rapidly increased to the peak in 10 s, which accelerated the formation of ∆pH during high light phases. However, such rapid activation of CEF did not appear by transitioning from 59 to 1809 μmol photons m^−2^ s^−1^ (Figure 3C), resulting in an insufficient formation of ∆pH (Figure 5A), the over-reduction in PSI (Figure 1C), and FL-induced PSI photoinhibition (Figure 4A). This is also consistent with previous studies in the delaying activation of CEF when exposed to FL in some environmental stresses such as heat and drought stress [13,14]. Therefore, a high level of background high light irradiance altered the CEF activation pattern in FL, which might affect the kinetics of ∆pH formation and the extent of PSI photoinhibition.

## 5. Conclusions

In this study, we examined the response of PSI to FL conditions with the same low light and different intensities of high light. We found that extreme high light could aggravate the PSI over-reduction and thus accelerate FL-induced PSI photoinhibition, which was mainly caused by insufficient ∆pH formation rather than the operation of LEF. When plants are inevitably faced with FL conditions, decreasing the intensity of high light is crucial for diminishing FL-induced PSI photoinhibition. For these reasons, we here provide insight into FL-induced PSI photoinhibition and strategies for crop cultivation under natural field FL conditions.

## Figures and Tables

**Figure 1 antioxidants-11-02325-f001:**
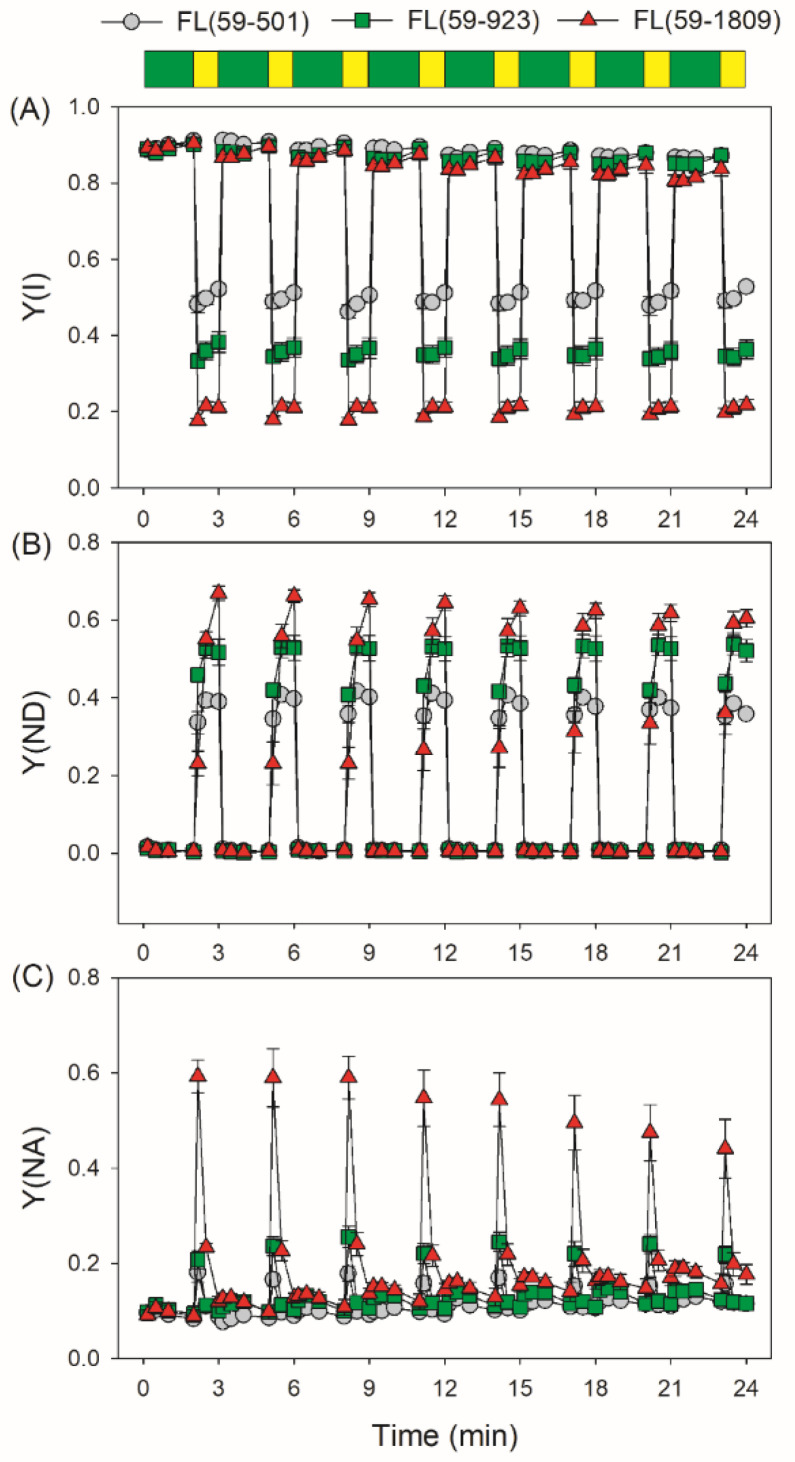
Changes in the PSI parameters under different fluctuating light conditions. After light adaptation at 1455 μmol photons m^−2^ s^−1^ for 5 min to activate photosynthetic electron sinks, leaves were exposed to fluctuating light alternating between low (59 μmol photons m^−2^ s^−1^, 2 min) and high light (501, 923 or 1809 μmol photons m^−2^ s^−1^, 1 min). The PSI parameters were recorded in low light (10 s, 30 s, 60 s and 120 s) and high light (10 s, 30 s and 60 s). (**A**) Y(I), the quantum yield of PSI photochemistry; (**B**) Y(ND), the quantum yield of PSI non-photochemical energy dissipation due to donor-side limitation; and (**C**) Y(NA), the quantum yield of PSI non-photochemical energy dissipation due to acceptor-side limitation. Data are shown as the means ± SE (*n* = 5). Green bars indicate low light (59 μmol photons m^−2^ s^−1^); yellow bars indicate high light (501, 923 or 1809 μmol photons m^−2^ s^−1^).

**Figure 2 antioxidants-11-02325-f002:**
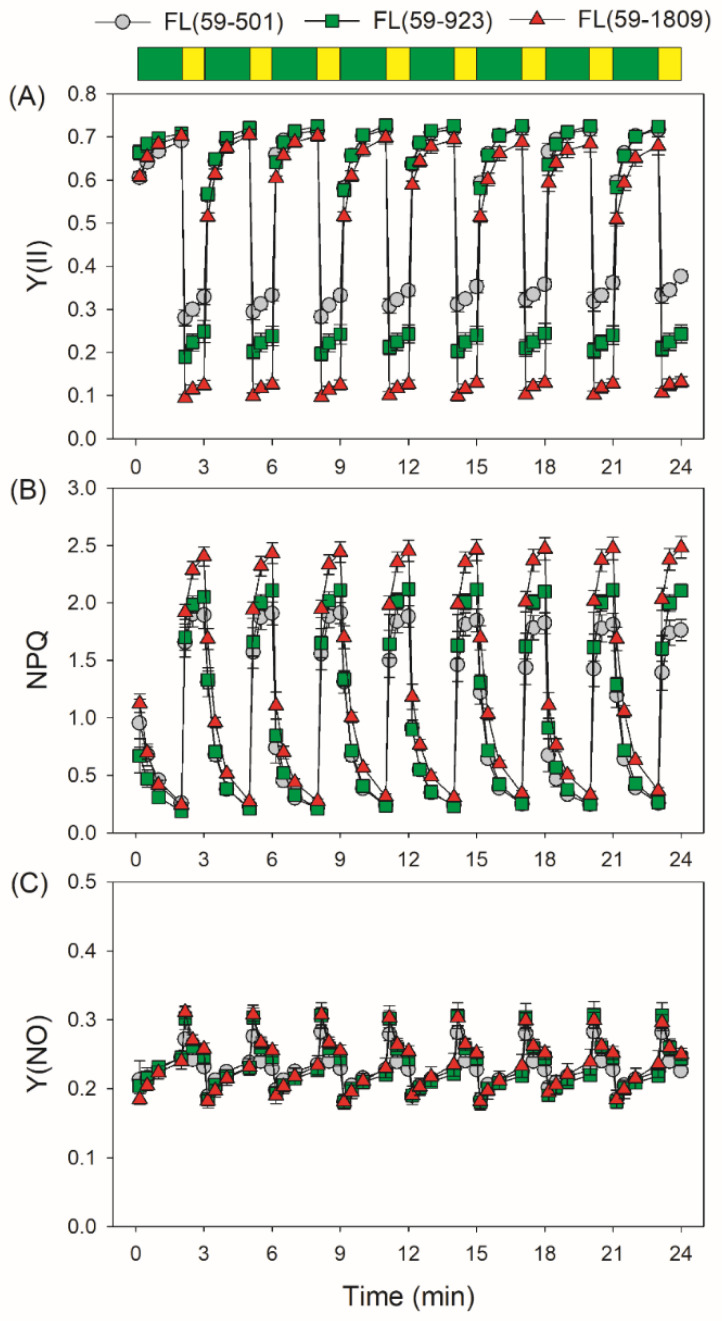
Changes in the PSII parameters under different fluctuating light conditions. After light adaptation at 1455 μmol photons m^−2^ s^−1^ for 5 min to activate photosynthetic electron sinks, leaves were exposed to fluctuating light alternating between low (59 μmol photons m^−2^ s^−1^, 2 min) and high light (501, 923 or 1809 μmol photons m^−2^ s^−1^, 1 min). The PSII parameters were recorded in low light (10 s, 30 s, 60 s and 120 s) and high light (10 s, 30 s and 60 s). (**A**) Y(II), the effective quantum yield of PSII photochemistry; (**B**) NPQ, non-photochemical quenching in PSII; and (**C**) Y(NO), the quantum yield of non-regulatory energy dissipation in PSII. Data are shown as the means ± SE (*n* = 5). Green bars indicate low light (59 μmol photons m^−2^ s^−1^); yellow bars indicate high light (501, 923 or 1809 μmol photons m^−2^ s^−1^).

**Figure 3 antioxidants-11-02325-f003:**
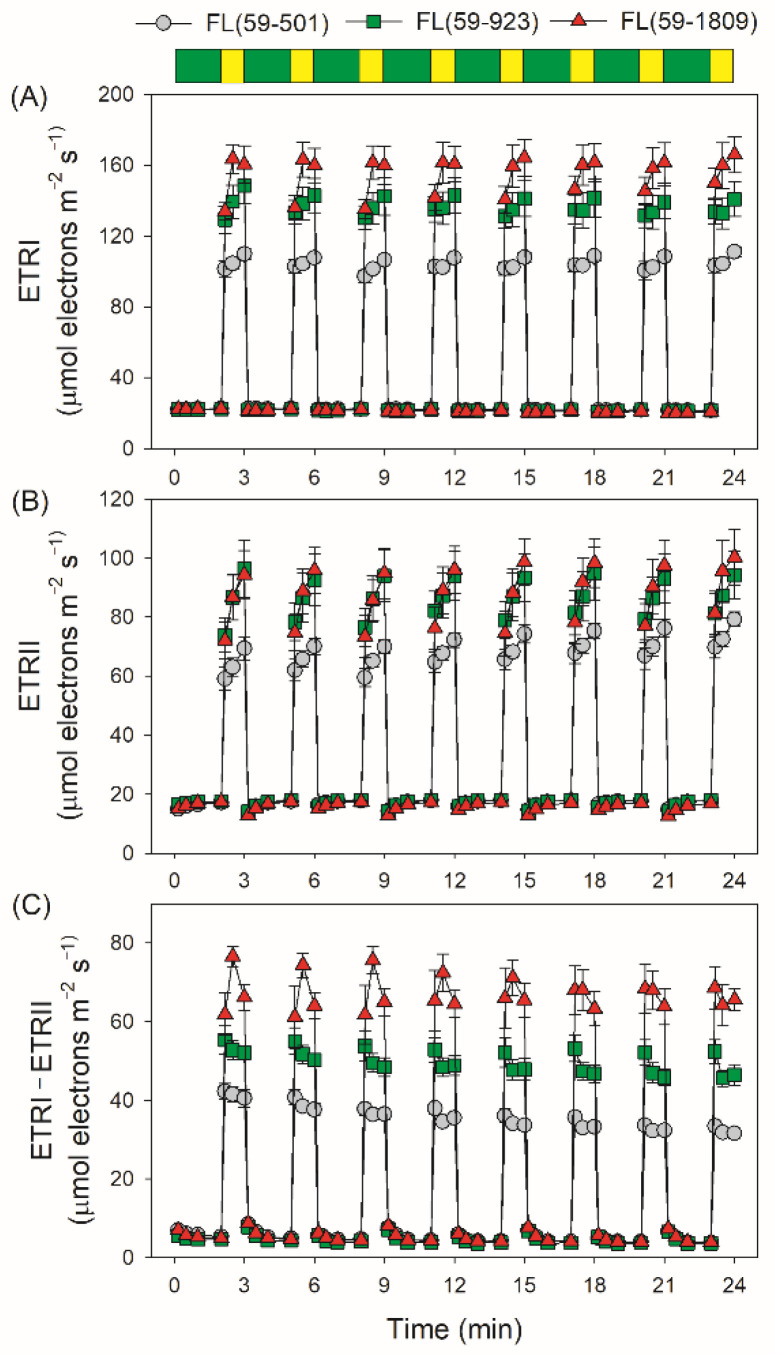
Changes in the electron transport rates under different fluctuating light conditions. After light adaptation at 1455 μmol photons m^−2^ s^−1^ for 5 min to activate photosynthetic electron sinks, leaves were exposed to fluctuating light alternating between low (59 μmol photons m^−2^ s^−1^, 2 min) and high light (501, 923 or 1809 μmol photons m^−2^ s^−1^, 1 min). The ETR values were calculated in low light (10 s, 30 s, 60 s and 120 s) and high light (10 s, 30 s and 60 s). (**A**) ETRI, ETR through PSI; (**B**) ERTII, ETR through PSII; (**C**) ETRI−ETRII, estimated rate of cyclic electron flow. Data are shown as means ± SE (*n* = 5). Green bars indicate low light (59 μmol photons m^−2^ s^−1^); yellow bars indicate high light (501, 923 or 1809 μmol photons m^−2^ s^−1^).

**Figure 4 antioxidants-11-02325-f004:**
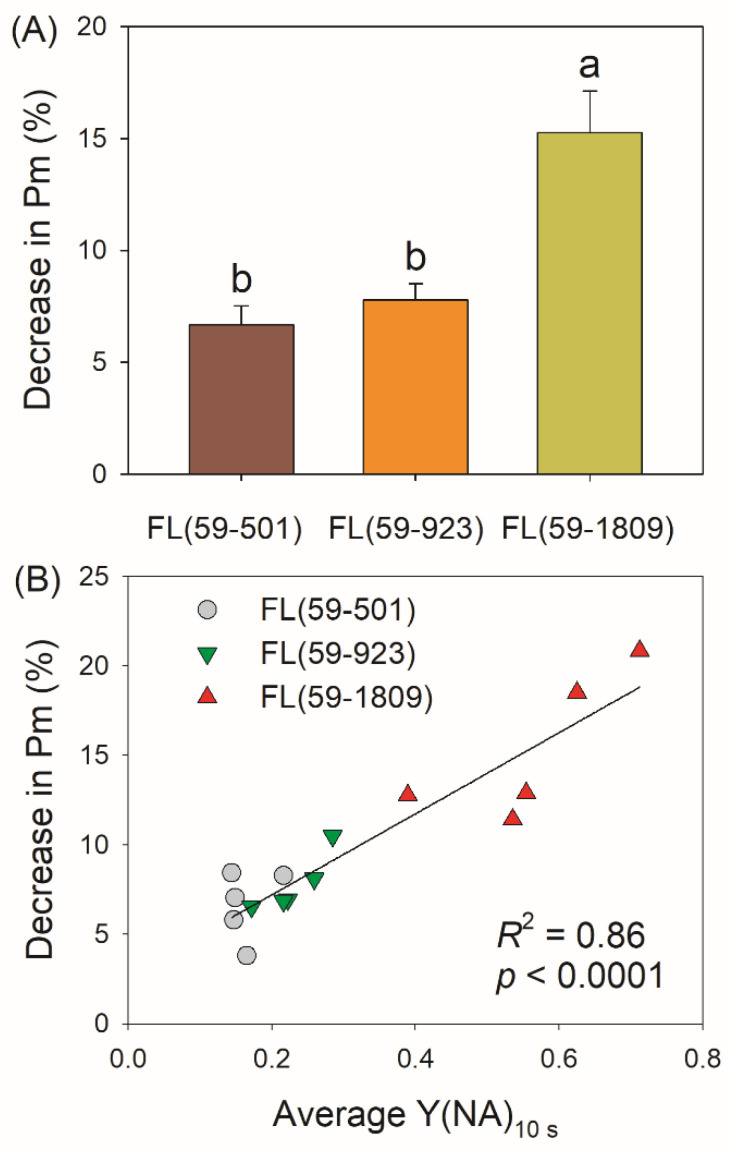
Photoinhibition under different fluctuating light conditions. (**A**) The decrease in the maximum photo-oxidizable P700 (*P_m_*). Data are shown as means ± SE (*n* = 5). Significant differences among different FL treatments are examined by Tukey–Kramer multiple comparison tests (*p* < 0.05). (**B**) The relationship between the decrease in *P_m_* and Y(NA)_10 s_. Y(NA)_10 s_ indicates the average value of Y(NA) within the first 10 s after switching to high light. Each plot represents the data of an individual leaf. Different letters (a and b) indicate significant differences between different treatments.

**Figure 5 antioxidants-11-02325-f005:**
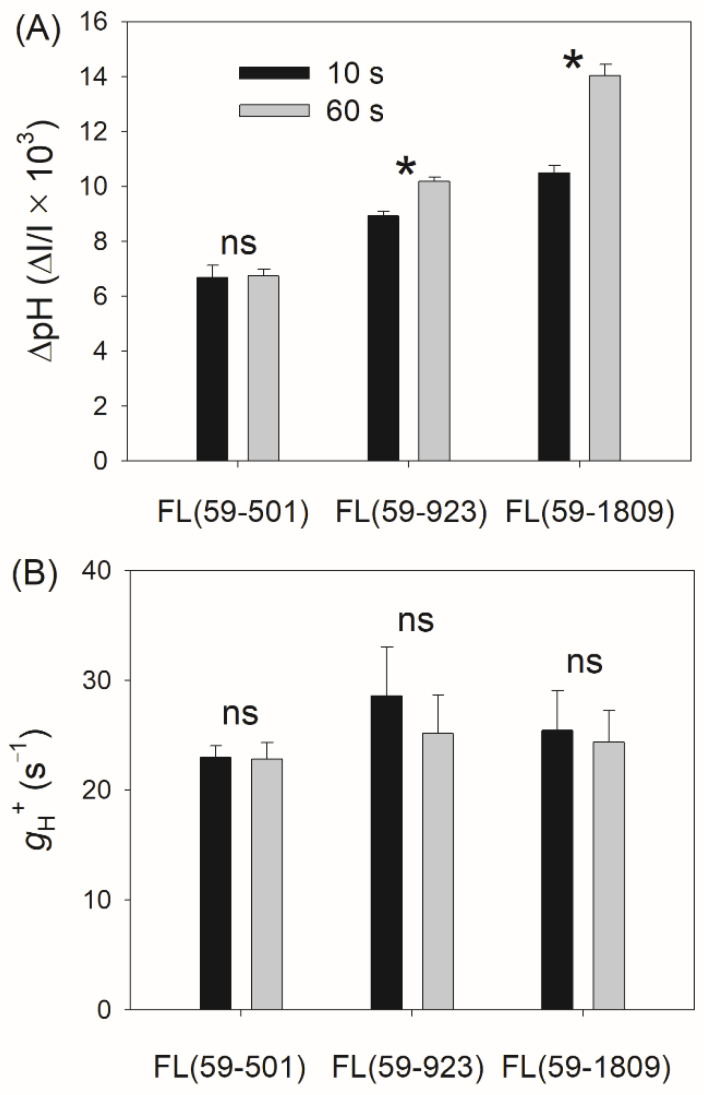
Changes in the proton gradient (∆pH) across the thylakoid membranes (**A**) and chloroplast ATP synthase activity (*g*_H_^+^) (**B**) under different fluctuating light conditions. ∆pH and *g*_H_^+^ were measured after the transition from 59 to 501, 923 or 1809 μmol photons m^−2^ s^−1^ for 10 s and 60 s. Data are shown as means ± SE (*n* = 5). The asterisk (*) indicates a significant difference between 10 s and 60 s, “ns” represents no significant difference.

**Figure 6 antioxidants-11-02325-f006:**
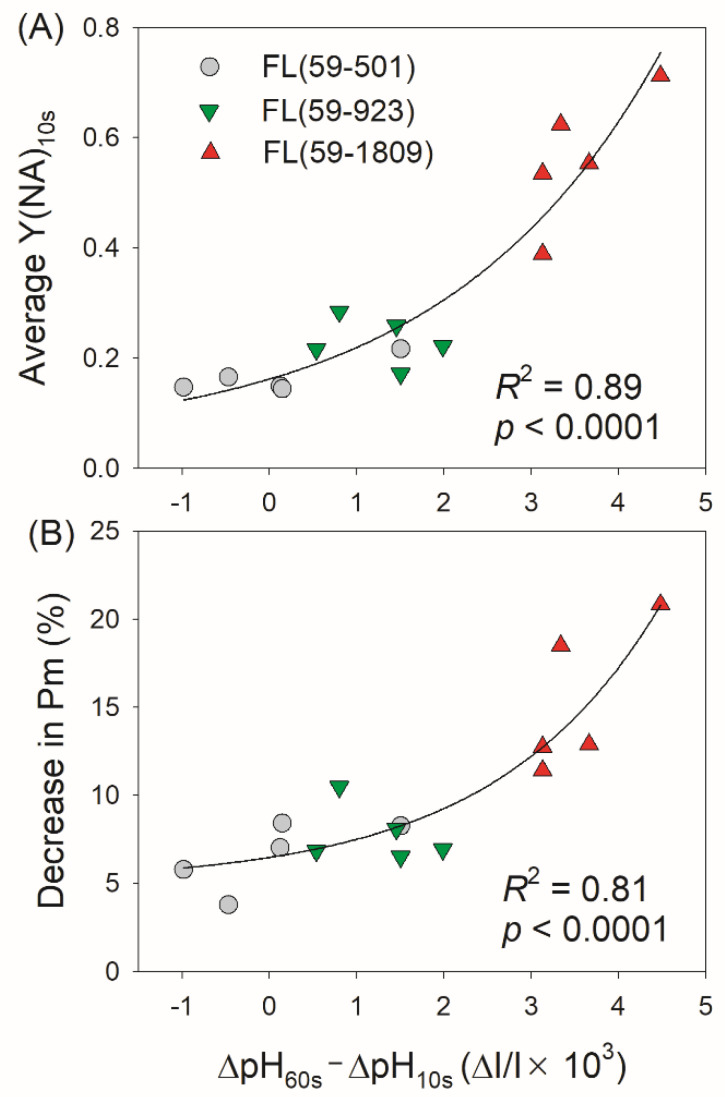
Effects of the delaying formation of ∆pH (∆pH_60 s_ − ∆pH_10 s_) on PSI redox state (**A**) and PSI photoinhibition (**B**) under different fluctuating light conditions. Each plot represents the data of an individual leaf. Exponential growth model (*y* = *y*_0_ + *ae^bx^*) was used to fit the relationships.

## Data Availability

The data presented in this study are available on request from the corresponding author.

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
