# Peer review of "Photoinhibition of Photosystem I Induced by Different Intensities of Fluctuating Light Is Determined by the Kinetics of ∆pH Formation Rather Than Linear Electron Flow"

_antioxidants, 2022, doi:10.3390/antiox11122325_

Round 1
Reviewer 1 Report
In the mentioned manuscript, attention is focused on the study of the influence of lighting changes on fluorescence parameters. The result of the experiment and its further use is not entirely evident from the manuscript. The results are in the field of theoretical basic research. From this point of view, it is an interesting topic. Tomato was used as a model plant. In the text, only the age of the plants is given, perhaps it would be appropriate to add the development phase according to the BBCH. What leaves were added to the measurement? The term physiologically mature leaf is a somewhat broad term, therefore it is necessary to add the developmental phase as well. Furthermore, it would be necessary to add a more detailed description of the soil. The general designation of humus soil is insufficient. Please clarify the indication of 40% solar radiation. The results are sleepy enough. It is a pity that graphs 1-3 are somewhat less clear, so it would be advisable to correct them. The discussion is rather descriptive. I recommend editing it.
Author Response
Thanks a lot for these important comments.
1. The developmental stage of plants and the developmental phase of leaves were described in the materials and methods (please see lines 112-113). After cultivating for a month, mature but not senescent leaves with the maximum photosynthetic capacity of the whole plant in elongation stage were used for photosynthetic measurements.
2. The nitrogen content of the humus soil was measured and provided in the revised version (please see line 110). All plants were grown with humus soil with the initial soil N content of 2.1 mg/g.
3. The light condition was described in more detail (please see lines 106-110). The light condition was controlled by a non-woven textile cover over them, reducing the photosynthetic active radiation (PAR) to approximately 40% of the sunlight (measured by a Li-1400 datalogger, Li-Cor Biosciences, Lincoln, NE, USA). As a result, the maximum light intensity exposed to leaves at noon was approximately 800 μmol photons m−2 s−1.
4. The section Results has been described in more detail.
5. The legends of Figures 1-3 have been revised.
6. The section Discussion has been checked to avoid descriptive.
Reviewer 2 Report
The manuscript of Shi et al. deals with the characterization of photoinhibitory damage of the Photosystem I (PSI) complex under fluctuating light. PSI photoinhibition, induced either by continuous or fluctuating light is a well-known and well characterized phenomenon. The current manuscript targets two aspects of this effect. One is the role of light intensity, which according to the authors has not yet been investigated. Although it is quite unlikely that there were no previous investigations on the light intensity dependence of PSI photoinhibition it is not impossible since based on our knowledge of light-induced damage of other biological systems, e.g. Photosystem II (PSII), it is a trivial assumption that the extent of light damage increases with light intensity if no other parameters of illumination (wavelength, or periodicity) are changed. In agreement with this trivial assumption it was indeed shown in the current manuscript that the increase of light intensity induces increased PSI photodamage. The other particular question was if PSI photodamage is related to the rate of linear electron flow (LEF, through PSII) or to pH gradient formation across the thylakoid membrane. From the observation that LEF saturated at a lower light intensity than PSI photodamage together with the observation that at high light intensity the formation of pH gradient takes longer time than at lower light intensities it was concluded that the important factor is the kinetics of pH gradient formation, i.e. “photoinhibition is more related to the kinetics pH gradient formation than to LEF”. This conclusion, however, is based on the apparent correlation of acceptor side limitation of PSI (Y(NA)) detected 10s after the onset high light and of pH gradient detected 60s after the onset of high light, which are not comparable quantities. Therefore, the experimental data do not support the statement on the role of pH gradient formation kinetics.
Taken together, the minor significance of the trivial light intensity dependence of PSI photodamage, and the unjustified correlation between PSI acceptor side limitation (PSI activity loss) with pH gradient formation kinetics the manuscript needs revision in the light of the comments below.
Problems:
1, There is a clear logical mistake in plotting Y(NA) at 10 s as a function of pH gradient difference between 60 and 10s in Fig. 6A. There is no way that Y(NA) measured at 10 s could be influenced by an event (pH gradient at 60s) that occurs 50 s later! Since this figure is the basis of the conclusion that the delay in the formation of pH gradient at higher light intensities would be the cause of increased PSI damage it has to be corrected. Either Y(NA)10s should be plotted as a function of pH gradient at 10s, or the average of Y(NA) for the 10 to 60s period should be plotted as a function of the average pH gradient in the 10 to 60 s period.
2, It is also unclear how the statistical analysis shown in Fig. 6 (A and B) was done. What was the fitting function, how were the R2 and P values calculated?
3, Fig. 6B shows the plot of Pm loss as a function of the pH gradient difference. However, the apparent correlation does not prove a cause-effect relationship here. Both parameters depend on light intensity, therefore, it is quite possible that high light induced loss of Pm and delay of the pH gradient buildup occur independently from each other, and the latter effect is not the cause of former one. According to Fig. 5A not only the difference of the pH gradient (between 10 and 60 s), but the absolute values (both at 10 and 60s) increase with light intensity. From that it follows that plotting the loss of Pm as a function of the absolute pH gradient values (at 10 s, but also at 60s or the average of 10 and 60s) would give a nice correlation, perhaps even better than the presented one. This of course would have just the opposite meaning (larger the pH gradient larger the PSI photodamage), which has to be verified. Therefore, the plot of Pm loss as a function of the absolute pH gradient values should also be presented and discussed.
4, PSI photoinhibition by FL is well characterized and reasonably well understood phenomenon, as shown by the cited papers/reviews in the manuscript. Therefore, the statement in the Abstract (l.16-17) that the effects of FL on PSI redox state and PSI photoinhibition are little known is not correct.
5, line 175. Y(NO) describes the combined pathways of radiative and non-radiative deexcitation reactions, which do not lead to photochemical energy conversion and are not involving the NPQ-mechanism. Why would it reflect PSII over-reduction? It should be explained.
5, Language problems:
line 84. shade-adapted instead of shade-establishing
line 167 and several other places: use intensity instead of magnitude for high light
line 173. non-regulated instead non-regulatory
line 244. What does the „rangeability” of FL means?
line 279. tought instead of toughted
line 289. There is a ca. 10% loss of Y(II) after 8 FL pulses, which shows a clear inhibition of PSII quantum yield (Fig.2A), so it is not correct to say that FL does not induce PSII photoinhibition at high light intensity.
line 306-307. Unclear meaning, no statement in the sentence.
line 312. Fig 4A does not show pH gradient values.
Author Response
Thanks a lot for these important comments.
1. The reviewer pointed out that either Y(NA)10s should be plotted as a function of pH gradient at 10s, or the average of Y(NA) for the 10 to 60s period should be plotted as a function of the average pH gradient in the 10 to 60 s period. According to this suggestion, we examined the relationship between pH gradient and Y(NA) after transition from low to high light for 10 s. However, the new figure was far away from our conclusion. Therefore, we did not add this new figure in the revised version due to the following reasons. First, the low Y(NA) in 60 s under a higher light needs a higher pH gradient, which was confirmed by Figure 5A. Secondly, after transition from low to high light, the performance of Y(NA) is more related to the relative pH gradient rather than the absolute pH gradient. For example, tomato leaves generated a higher pH gradient in 10 s after transition from 59 to 1809 μmol photons m−2 s−1 than transition from 59 to 501 μmol photons m−2 s−1 (Figure 5A) However, the transient high Y(NA)10 s was observed in FL (59-1809) but disappeared in FL (59-501) (Figure 1C). Therefore, we plotted Y(NA) at 10 s and the decrease in Pm as a function of pH gradient difference between 60 and 10s in Fig. 6. The results indicated that the greatest PSI photoinhibition induced by FL (59-1809) was caused by the insufficient pH gradient in 10 s.
2. The statistical analysis in Fig. 6 was described in more detail.
3. The related sentence in section Abstract has been rewritten.
4. We explained why Y(NO) reflects PSII over-reduction in more detail.
5. Other language problems have been revised.
Round 2
Reviewer 1 Report
The authors submitted a revised version of the manuscript. The revised manuscript is of a better quality than the previous version. The authors accepted the comments and comments of the reviewer. They gave an explanation for the changes.
Author Response
Thanks a lot for your review.
Reviewer 2 Report
The revised manuscript of Shi et al. corrected the language and other minor problems.
However, for the main point, which is the incorrect plotting of data in Fig. 6. they answered the following: „we examined the relationship between pH gradient and Y(NA) after transition from low to high light for 10 s. However, the new figure was far away from our conclusion. Therefore, we did not add this new figure in the revised version”!
I would like to draw the attention of the authors that in science the conclusions are based on the data, and not the data (or their presentation) are adjusted to a preconception. If the plot of Y(NA) at 10s as a function of the pH gradient at 10s, or of the average of Y(NA) in the 10-60s period as a function of the average pH gradient in the same period gives a result which contradict to the conclusion then the conclusion should be changed instead of hiding the data.
Although the authors attempted the explain in their answer that their original conclusion is still correct, the obvious contradiction between the correct plot of the data (Y(NA) as a function of pH gradient measured in the same time period) cannot be ignored since this would be the essence of the work (the light intensity dependence is minor, trivial issue).
Therefore, the revised manuscript is unsuitable for publication and and should be rejected.
Author Response
I think the Reviewer might misunderstand us. First, our conclusions in science are based on the data rather than the relationship between data. Secondly, the redox state of Y(I), especially the value of Y(NA), is mainly determined by the relative pH gradient rather than absolute value. Light responses of pH gradient and Y(NA) has been investigated in many previous studies. In general, a high pH gradient is required to maintain the steady-state Y(NA) under a higher light intensity. Consistently, with the increase in light intensity (from 501 to 1809 µE), the steady-state value of Y(NA) was approximately 0.1, and the same value of Y(NA) required a much higher pH gradient at 1809 µE than at 501 µE. Under a given light intensity, an insufficient pH gradient results in PSI over-reduction, as indicated by the phenotypes of CEF mutants. Therefore, PSI over-reduction occurs only when the incident pH gradient is lower than the steady-state pH gradient, pointing out that the difference between incident pH gradient and steady-state pH gradient is the primary determinant of PSI redox state after transition from low to high light. Based on this pattern, we analyzed the relationship between Y(NA)10s and ∆pH60s – ∆pH10s to understand why the transient PSI over-reduction in 10 s just occurred after transition from 59 to 1809 µE. We believe this analysis is scientific and finely supports our conclusion.
Round 3
Reviewer 2 Report
The second revision of the manuscript of Shi et al. does not contain any modification, and does not include the figure, which was requested in the first review.
The authors still insists on showing the incorrect version of Fig. 6. in which a dependent variable (Y(NA) measured at 10 s) is shown as a function of an independent variable (pH gradient difference) measured at a later time period 10-60 s as the dependent variable was obtained. Their argument given in the answer for the validity of this approach is not acceptable. There is no way, according to the known rules of natural sciences, that a quantity (pH gradient difference) which develops at a later time (from 10 to 60 s) could influence another quantity (Y(NA)) which is measured at an earlier time (at 10 s). A time travel would be needed for that a later event could influence an earlier event.
That is why I suggested that the plot should show either Y(NA) at 10 s as a function of the pH gradient at 10 s, i.e. both measured at the same time, or the average of Y(NA) measured between 10 to 60 s, plotted as a function of the average pH gradient measured also between 10 to 60 s, i.e. both quantities reflect averages from the same time period. In the first revision the authors answered that they made the modified plot, but it did not support their conclusion, so they decided do not show it. This is not how science works. The only conclusion is that their original idea is wrong, since not supported by the correctly plotted experimental data.
Therefore, the manuscript is not acceptable in the present form.
Author Response
In this revision, we have added the relationship between pH gradient and Y(NA) after transition from low to high light for 10 s and 60 s (please see Figure S1). The data also indicated that the transient PSI over-reduction in FL (59-1809) was tightly correlated to an insufficient pH gradient.